# Comparing Computational Peritoneal Dialysis Models in Pigs and Patients

**DOI:** 10.3390/toxins17070329

**Published:** 2025-06-28

**Authors:** Sangita Swapnasrita, Joost C. de Vries, Joanna Stachowska-Piętka, Carl M Öberg, Karin G. F. Gerritsen, Aurélie Carlier

**Affiliations:** 1MERLN Institute for Regenerative Medicine, Maastricht University, Universiteitssingel 40, 6229 ER Maastricht, The Netherlands; s.swapnasrita@maastrichtuniversity.nl; 2Department of Nephrology and Hypertension, University Medical Center Utrecht, Heidelberglaan 100, 3584 CX Utrecht, The Netherlands; j.c.devries-34@umcutrecht.nl; 3Nalecz Institute of Biocybernetics and Biomedical Engineering, Polish Academy of Sciences, Ks. Trojdena 4, 02-109 Warsaw, Poland; jstachowska@ibib.waw.pl; 4Department of Clinical Sciences Lund, Division of Nephrology, Skåne University Hospital, Lund University, 221 85 Lund, Sweden; carl.oberg@med.lu.se

**Keywords:** peritoneal dialysis, three-pore model, uremic toxin, mathematical modeling, personalization

## Abstract

Computational models of peritoneal dialysis (PD) are increasingly useful for optimizing treatment in patients with kidney disease requiring dialysis (KDRD). However, although several mathematical models have been developed in the past few decades, a direct comparison of the models’ accuracy with respect to predicting in vivo data is needed to further create robust personalized models. Here, we used a dataset obtained in a previous in vivo experimental model of PD in pigs (23 sessions of 4 h 2 L dwells in four pigs) and humans (20 sessions in 20 patients) to compare six computational models of PD: the Graff model (UGM), the three-pore model (TPM), the Garred model (GM), and the Waniewski model (WM), as well as two variations of these (UGM-18, SWM). We conducted this comparison to predict the dialysate concentrations of key uremic toxins and electrolytes (four in humans) throughout a 4 h dwell. The model predictions can provide insight into inter-individual differences in ultrafiltration, which are critical for tailoring PD regimens in KDRD. While TPM offered improved physiological reality, its computational cost suggests a trade-off between model complexity and clinical applicability for real-time or portable kidney support systems. In future applications, such models could provide adaptive PD regimens for tailored care based on patient-specific toxin kinetics and fluid dynamics.

## 1. Introduction

Dialysis is a life-sustaining therapy that removes waste products and excess fluid from the blood in patients with kidney failure to partly replace the kidney function. Hemodialysis (HD) is the process of filtering out toxins from the blood by flowing it through an extracorporeal circuit, while peritoneal dialysis (PD) is the process of removing excess water and uremic toxins by introducing a dialysis fluid with osmotic agents to the abdominal cavity of patients. PD offers several benefits over HD, including the continuous and gradual removal of waste products, which allows for the avoidance of the sharp fluctuations in solute and fluid levels typical of HD’s intermittent ‘saw-tooth’ pattern. PD does not require vascular access, thereby reducing associated complications, and can easily be performed at home, promoting greater patient autonomy and quality of life. PD better preserves residual kidney function, which is associated with improved patient outcomes, and is generally less expensive. Despite these benefits, PD faces important limitations. Technique survival is limited, with a median duration of approximately 3.7 years, due to complications such as recurrent peritonitis, catheter malfunction, or long-term damage to the peritoneal membrane from exposure to the high-glucose dialysate [1,2]. Additionally, PD is less effective in clearing toxins compared to HD [3,4]. As a result, many patients eventually have to transition to HD due to technique failure or inadequate toxin clearance as residual kidney function declines.

Over the >60 years’ progress in understanding the PD process, PD mathematical models have grown in complexity [5,6,7,8,9,10,11,12,13,14,15,16]. Mathematical modeling is a good way of combining physics with clinical knowledge. It facilitates the quantification of solute and fluid transport necessary for optimizing the PD process. PD models can also help to quantify the structure and physiological state of peritoneal tissue and its transport characteristics.

The PD models of Kallen [17], Miller [18], and Henderson [19] are two-compartmental models, which consider the peritoneal membrane to be homogeneously porous and the solute transfer across the membrane to be purely diffusive. However, as our understanding of the peritoneal membrane has increased, we now know that there is also convective solute transfer and that solutes are transferred depending on their size and charge, thus indicating a heteroporous network. Such a heteroporous network was incorporated into the simplified models of Garred [20] and Waniewski [21,22]. We also know that there are aquaporin channels for water transfer [23] and that the lymphatic system is important for the removal of fluid and macromolecules [24,25,26,27,28]. Consequently, recent PD models like the solute-specific models of Graff and Fugleberg [23,24,25,26,27,28], the three-pore model (TPM) of Öberg [10], the empirical model of Gotch [29], and the distributed model of Flessner [30,31] have incorporated this complexity, underscoring the importance of diffusion, convection and lymphatics.

However, a rigorous comparison of the accuracy of various models is needed, since an accurate model may contribute to the optimization and personalization of this life-saving therapy and may be used to optimally design PD innovations that are currently under development, such as sorbent-assisted continuous flow PD [32,33]. A robust evaluation framework should encompass (1) a defined set of benchmarks with which the model’s performance is tested, (2) metrics for measuring and comparing the models to identify strengths and areas for improvement, and (3) avenues for enhancing the PD procedure.

In this paper, we focus on comparing the accuracy of four models (plus two variations), both in humans and pigs, since pigs are commonly used for the preclinical validation of PD innovations [32,34]. This is a benchmarking modeling study aimed at evaluating how accurately the models predict the dialysate concentrations and mass transfer area coefficients (MTACs) of six solutes (urea, creatinine, phosphate, potassium, glucose, and sodium) in pigs and four solutes (urea, creatinine, glucose, and sodium) in humans. The MTAC, which represents the maximal diffusive clearance, reflects the capacity for diffusion across the peritoneal membrane and can be calculated based on solute transport dynamics. We show that some models are only accurate for specific solutes, while other models can correctly predict multiple solute concentrations. To the best of our knowledge, this is the first time that TPM has been applied to pigs. As such, this work gives us crucial insights into which model could be utilized to accurately represent the dynamic interplay of solute transport, thereby paving the way for personalized dialysis care.

## 2. Methods

We compared four existing models (Table 1) and determined which computational model of PD predicted the in vivo and patient data with the least error. The four models were chosen because of their widespread use in PD modeling or ease of use in clinical practice. Since the transport of a solute is determined by its hydrodynamic size, these models are solute-specific. However, we were also interested in a generalized model, a model that can be applied at once to multiple solutes and provide an accurate determination of peritoneal mass transfer per subject. Therefore, we also included two generalized models (UGM-18 and SWM), which are variations of two of the four models [35].

We did not include the distributed model [31] in our analysis on account of its complexity. The distributed model focuses on the peritoneal transport in the tissue itself based on solute diffusivity in tissue, capillary permeability, and surface area. In particular, the distributed model is of importance when the solute transport from the peritoneal membrane into the tissue needs to be investigated, which is not within the scope of this work.

### 2.1. Data Collection

All the data used to compare the models were procured retrospectively. As this study was conducted retrospectively, the set of solutes analyzed in pigs and humans reflects the measurements available from prior experimental protocols: six solutes for pigs (urea, creatinine, phosphate, potassium, glucose, and sodium) and four solutes for humans (urea, creatinine, glucose, and sodium). Consequently, not all solutes were available for modelling in both species.

For the pig data, the experimental procedure can be found elsewhere [36]. Briefly, after an overnight dwell, 2 L of Physioneal 35 1.36% glucose (*n* = 19 in 3 pigs) or 2.27% glucose (*n* = 4 in 2 pigs) was instilled, and PD drain samples were collected at 0, 10, 20, 30, 60, 120, 180, and 240 min for measurement of urea, creatinine, sodium, phosphate, glucose and potassium. After 240 min, a complete drain was performed, followed by flushing the peritoneal cavity with 1 L of fresh Physioneal 35 1.36% glucose to determine the residual volume based on the dilution of the total protein or albumin concentration in the drained fluid (see Section 2.3.1). Blood solute measurements were taken at *t* = 0, 120, and 240 min. The instilled and drained intraperitoneal volumes were measured, along with blue dextran measurements for intermediate volume data. In short, blue dextran (Sigma Aldrich, Taufkirchen, Germany; D5751-10g, 2000 kDa) was dissolved in fresh Physioneal 35 1.36% and added to the dialysate for the fresh fill at the start of the standard peritoneal assessment to achieve a concentration of 0.49 ± 0.14 g/L (*n* = 21). Samples for blue dextran measurements were taken from the bag prior to instillation, and intraperitoneal samples were taken at t = 0, 10, 20, 30, 60, 120, 180, and 240 min. Absorbance was measured at 620 nm, and sample blue dextran concentrations were calculated. Three datasets were excluded due to inaccurate residual volume measurement.

For the human data, we retrospectively included 20 patients on PD who underwent a regular peritoneal equilibration test (PET, using Physioneal 1.36% (*n* = 1), 2.27% (*n* = 1) and 3.86% (*n* = 18)) at any time during their treatment period in the University Medical Centre Utrecht (Utrecht, The Netherlands). Patients were excluded if they had established pathology that may interfere with the PET (e.g., active peritonitis, catheter dysfunction, or other relevant abdominal pathology) or if there was missing data for the PET. In all patients for whom multiple PETs were performed, data from the most recent PET was extracted. An ethics board waiver for this study was obtained from the local ethical committee, and informed consent was signed with the patient in accordance with local (privacy) laws and guidelines, prior to inclusion in the study (if applicable). For human data, we used urea, creatinine, sodium, and glucose dialysate concentrations that were measured at time intervals of 20, 120, and 240 min. Total protein was measured from the following four time points: overnight dwell drain, fresh fill, dwell drain of PET, and new fresh fill. As initial dialysate concentrations of the fresh dwell, we used the concentrations reported by the manufacturer (zero for urea and creatinine, 132 mmol/L for sodium, and 75.6, 126.1, or 214 mmol/L for glucose, according to the PET solution used). Blood solute measurements were performed in the middle of the dwell (*t* = 120 min). The instilled and drained intraperitoneal volumes were measured. Eleven datasets were excluded because of missing or imprecise total protein measurements.

### 2.2. Model Descriptions

Model descriptions for all six models can be found in Table 1 and in the Appendix A.

### 2.3. Model Fitting Procedure

In the realm of PD computational modeling, particularly in relation to predicting solute concentration (y), two distinct methodologies emerge. UGM, UGM-18, TPM, and WM employ an iterative process wherein MTACs are adjusted within a predictive equation to minimize the discrepancy, measured by the root mean square error, between the model’s dialysate solute concentration predictions and the experimental data, $\;y_{experimental}$. This approach essentially fine-tunes the model parameters through an optimization process, in which the goal is to harmonize the model’s outcomes with observed values as closely as possible without direct derivation from the experimental results. Finally, in addition to the iterative process for MTACs, TPM includes a dynamic fluid model, which implies that we also minimize the volume-related error, measured via the root mean square error, between the model’s intraperitoneal volume and measured intraperitoneal volume.

Conversely, the second method (GM and SWM) adopts a more direct approach, leveraging $y_{experimental}$ to calculate the values of MTACs explicitly. This methodology avoids the iterative error-minimization strategy of the first model in favor of utilizing observed data to directly ascertain the parameters’ values. Here, experimental observations serve not as a benchmark for optimization but as the foundational basis from which the model parameters are directly derived.

#### 2.3.1. Fitted Models: UGM and UGM-18

The steps for fitting the dialysate concentration of sodium, potassium, creatinine, phosphate, glucose, and urea to determine the fitting parameters are outlined below and shown in Figure 1. For human data, only urea, creatinine, sodium, and glucose dialysate concentrations were fitted. We assumed that vasodilation of the peritoneal membrane in response to the dialysis fluid did not occur, and a stable mass transfer coefficient was maintained throughout the dwell session.

Load subject-specific data: The instilled and drained intraperitoneal volumes were taken from the data. The initial/final intraperitoneal volume was calculated as the sum of the instilled/drained volume and residual volumes. Residual volumes were calculated from the dilution of albumin for the pig data (or total protein if albumin was not available) or total protein for human data using the concentrations in the drained fluid and the new dwell just after instillment of the fresh dialysate. In PETs where no dextran was measured (all human PETs and 1 PET in pigs), the intraperitoneal volume, V, was linearly interpolated from the initial and final values. We know this may differ from the actual volumes, but since low glucose concentrations were used experimentally in the pigs, we assumed that a linear interpolation would not introduce many errors. We used the same interpolation for humans, where 3.86% glucose was mostly used, which could lead to an underestimation of the volume at the beginning of the dwell, leading to an underestimation of solute removal. However, for TPM, we used a dynamic fluid model, which we also fitted by using the following parameter: lymphatic flow rate, L. The plasma solute concentrations, $c_{p}$, values were taken from the data and interpolated between sampling points for pigs and assumed to be constant for humans.Initialize optimization parameters and bounds: To run the fitting program, the simulation was started with an initial estimation of the fitting parameter, and the bounds of the parameter were set to encompass the variability of the parameter found in the literature. For example, in UGM in pigs, 11 parameters were fitted (6 MTACs and 5 sieving coefficients (SiCo)—SiCo for glucose in UGM is 0). We would initialize $MTAC_{urea}$ as 10 mL/min and set the bounds of this parameter between 0 and 200 mL/min, which cover the known ranges of urea MTAC (only for TPM, UGM, and UGM-18).Minimize the root mean square error: Our aim is to determine which model could predict the dialysate concentration for humans and pigs. In order to compare the predictions for the four/six solutes across the models, we fitted model-specific parameters in each case to compare the normalized solute concentration error, $C_{error}$ (Equation (3)), between the predicted and measured dialysate concentrations.Steps 2–3 were repeated 10 times with different initializations (randomization of MTACs) to find the global minima. The minimization in step 3 is performed with Python 3.9 using Sequential Least Squares Programming (SLSQP). SLSQP is an optimization algorithm that is widely used for solving nonlinear optimization problems with both equality and inequality constraints. It employs a sequence of quadratic programming subproblems, optimizing a function by iteratively approximating it as a quadratic function and adjusting the variables to meet the constraints.Calculate the normalized solute-specific concentration error (SSE): We checked the error in concentration determination per solute along with the $C_{error}$ to assess both the individual solute fits and the overall accuracy of the model.

#### 2.3.2. Fitted Model: TPM

All steps were followed as outlined in Section 2.3.1. However, TPM includes a dynamic fluid model and a dynamic solute model. Thus, there is a minimization of error in the volume profile, as well as the solute profile. The volume error is minimized as follows:(1)$$V_{error}=\sqrt{\frac{\sum {\left(\frac{V_{predicted}-V_{measured}}{\max V_{measured}}\right)}^{2}}{nt}}$$
where nt is the number of time steps at which the volume is measured, and $V_{predicted/measured}$ is the predicted and measured intraperitoneal volume, respectively.

#### 2.3.3. Derived Models: GM and SWM

Subject-specific data were loaded as described above. The MTAC was calculated by reconfiguring Equation (S1.10) (in the Appendix A) to(2)$$MTAC=\frac{V_{mean}}{240}\times \log\left(\frac{V_{0}^{1-f}(c_{p,mean}-c_{d,0})}{V_{240}^{1-f}(c_{p,mean}-c_{d,240})}\right)$$
where f = 0 for GM and 0.5 for SWM. $V_{mean}$ is the mean intraperitoneal volume during the dwell. $c_{p,\;mean}$ is the mean recorded plasma concentration, and $c_{d,t}$ is the solute dialysate concentration at time *t*. For patient data, $V_{0}$ was replaced by $V_{20}$, and $c_{d,0}$ was replaced by $c_{d,20}$. $C_{error}$ and SSE were calculated for the derived MTAC values using Equations (3) and (4), respectively.

#### 2.3.4. Derived Model: WM

Subject-specific data were loaded as described above. Linear regression was performed to fit MTAC and SiCo for all solutes. $C_{error}$ and SSE were calculated for the derived MTAC values using Equations (3) and (4), respectively.
*Calculation of* $C_{error}$ *and SSE*

$C_{error}$ and SSE were calculated using the following formulas:(3)$$C_{error}=\sum_{all\;solutes}\sqrt{\frac{\sum_{t=0}^{t=end}{\left({\tilde{c}}_{d,\;predicted}-{\tilde{c}}_{d,measured}\right)}^{2}}{nt}}$$(4)$$SSE=\sqrt{\frac{\sum_{t=0}^{t=end}{\left({\tilde{c}}_{d,\;predicted}-{\tilde{c}}_{d,measured}\right)}^{2}}{nt}}$$
where ${\tilde{c}}_{d,predicted}$ and ${\tilde{c}}_{d,measured}$ are the normalized predicted and measured dialysate concentrations, respectively, and nt is the number of sample time points of the concentration. The solute concentrations were normalized according to their respective plasma solute concentrations at *t* = 0. For glucose, however, the concentration was normalized according to its dialysate concentration at *t* = 0 to ensure that all normalized values remained between 0 and 1.

The minimization criteria for TPM are slightly different from those of Equation (3), as follows:(5)$$Criteria=C_{error}+V_{error}$$

All models were implemented in Python 3.10 and are available on Github (https://github.com/carliercomputationallab/PD-comparison-paper.git (accessed on 25 June 2025)). All calculations were performed on a personal workspace equipped with Intel^®^ Xeon^®^ Gold 5415+, 2900 Mhz, 8 cores, and 16 logical processors.

### 2.4. Ultrafiltration Calculations

Measured ultrafiltration was calculated as follows:(6)$$UF_{measured}=\left(V_{drain}+V_{res,\;240}\right)-(V_{fill}+V_{res,\;0})\;$$
where $V_{drain}$ and $V_{fill}$ are the drained and infused volumes, respectively, and $V_{res}$ is the residual volume calculated from the albumin or total protein measurements.

## 3. Results

### 3.1. Concentration Predictions for Specific Subjects

Figure 2 shows the mean predicted dialysate concentration at different time points for each solute and model in pigs. For sodium, we see major differences between the models. Sodium sieving is characterized by a dip in dialysate sodium concentration during the early dwell, which is caused by the fast osmotic transfer of water into the abdominal cavity through aquaporins [37,38]. Both WM and TPM captured the initial dip in sodium concentration. In contrast, GM and SWM, which interpolate values without a volume model, failed to capture the dip entirely. UGM predicted an initial increase and could not capture the dip at all. UGM-18 predicted a dip in the sodium profile but was not able to predict the subsequent increase when sodium equilibrates across the peritoneum. These findings underscore the importance of accurately representing the early and prominent role of aquaporins in a PD model. Models like TPM, in which we included the dynamic fluid model, and WM, which incorporates a static fluid model, provide a better representation of sodium kinetics. Individual pig dialysate concentration profiles can be found in Appendix A.

Figure 3 shows the mean predicted dialysate concentration at different time points for each solute and model in humans. Like pigs, sodium profiles are not predicted accurately by models that do not include a fluid model, such as GM, SWM, UGM, and UGM-18. TPM and WM make reasonable predictions for the sodium dip. Individual human dialysate concentration profiles can be found in Appendix A.

### 3.2. Performance of Various Models of Peritoneal Dialysis

The six models were used to fit the dialysate solute concentration of the 20 dwell sessions in pigs and 9 dwell sessions in humans. The $C_{error}$ values (mean and standard deviation), given by either Equation (3) or Equation (5), for the six models are shown in Figure 4. In pigs, the accuracy was the highest for WM (lowest $C_{error}$ = 0.08 ± 0.05). TPM predicted parameters with reasonable accuracy overall (low $C_{error}$ of 0.13 ± 0.05). GM and SWM had comparable accuracy (0.17 ± 0.05 and 0.15 ± 0.05, respectively). The unified Graff model (with the best fitted parameters, UGM and the fully random model, UGM-18) had $C_{error}$ values of 0.13 ± 0.04 and 0.21 ± 0.09, respectively. TPM, GM, and SWM predicted all solutes’ concentrations with high accuracy (SSE < 0.05), except for potassium in the case of GM (Figure 5). Among all models, WM was the best-performing, and UGM and UGM-18 were the worst-performing. The phenomenon of sodium sieving could not be captured well by the many models, as the concentration of sodium was dependent on ultrafiltration, which is lumped in every model except TPM. This led to negative MTAC values (Figure 6 and Appendix A) in some models (WM, SWM, and GM), even though the SSE seemed to be low.

In humans, WM predicted parameters with the highest accuracy (WM: 0.00 ± 0.00). UGM and UGM-18 had similar $C_{error}$ values of 0.18 ± 0.06. TPM had a $C_{error}$ value of 0.19 ± 0.07. GM and SWM had comparable accuracies of 0.21 ± 0.07 and 0.23 ± 0.07, respectively. All models predicted most solute concentrations (except glucose) with great accuracy (SSE ≤ 0.05). Most models, except WM, predicted glucose with a high error rate ≥ 0.05, and TPM predicted urea with a SSE value of 0.06 (Figure 5).

WM and UGM-18 predicted solute concentrations with higher accuracy in humans than in pigs, while TPM, GM, UGM, and SWM predicted solute concentrations with lower accuracy in humans than pigs.

The computational time taken to execute one full fitting process independent of the number of fitted parameters is shown in Figure 4b. The SLSQP program used to minimize the objective function for TPM, UGM, and UGM-18 has a time complexity of *O*(*n*^3^), where *n* is the number of fitted parameters [39]. The largeness of the matrix decomposition employed (how many parameters are fitted; see Table 1) by SLSQP [40] resulted in increasing computational time for TPM, UGM, and UGM-18. The MTACs were derived from the measured concentrations at two time points for the GM and SWM models and thus required less computational time. WM, using a linear regression model, is also a relatively fast computational method. Mostly, the computational time was lower for humans than for pigs due to the lower number of solutes (and, thus, the number of MTACs to be fitted).

All fitted parameters (MTAC, fct (i.e., the relationship between diffusion and convection, which is only relevant to Graff models (dimensionless)), sieving coefficients, and lymphatic flow rates) can be found in Appendix A.

### 3.3. Population Average of Predicted MTACs in Pigs and Humans

In Figure 6, we see the population average of the predicted MTACs in pigs and humans. All models predict comparable MTACs for urea and creatinine, both in pigs and in humans. For urea, creatinine, and glucose, higher MTACs are noted in humans compared to pigs. WM could not reliably predict glucose MTAC in pigs.

For sodium, we see the maximum discrepancy within the models. For the linear regression model, WM, the predicted MTAC was mostly in the negative range. Negative sodium MTACs were observed for GM and SWM in pigs and humans. This can be attributed to the incorrect assumption of linear interpolation between the initial and final volumes.

For phosphate and potassium, WM predictions are higher than those of other models. We observed that the fitted and predicted MTACs in humans for creatinine, urea, and glucose in this study are generally higher than those found in the literature with TPM [10,11]. Due to the unavailability of human data for phosphate and potassium, we could not provide comparisons. Other fitted parameters, including the diffusion–convection ratios, fct, and sieving coefficients, SiCo, can be found in Appendix A.

### 3.4. Prediction of Ultrafiltration Volumes and Lymphatic Flow Rates in Pigs and Humans

We see in Figure 7 that the predicted ultrafiltration volume (with TPM) is very strongly correlated with the measured ultrafiltration values (Equation (6)), both for pigs and humans. The mean lymphatic flow rate (L) is slightly higher in pigs than in humans. $V_{err}$ (Equation (1)) values calculated in pigs and humans are similar.

## 4. Discussion

The growing demand for home-based dialysis has led to a rise in PD. In response, new PD devices (automated wearable artificial kidney (AWAK-PD) [33], Carry Life PD [41], and sorbent-assisted PD [32]) are being developed to make PD more accessible and efficient for patients. As these devices undergo testing and refinement, it is crucial to integrate modeling into device development to (1) reduce unnecessary animal testing and clinical trials, (2) make specialized improvements, and (3) perform virtual trials to pre-assess the validity and use case of such devices and determine the optimal operating conditions. What is currently lacking is a rigorous assessment of the conventional PD models and validation against both animal data and human data. Thus, in this study, we have chosen six models of varying physical and temporal complexity to benchmark against the experimental and clinical data. The experimental data are reused from previous studies to save time and resources and enhance the credibility of the prior research findings. In this benchmarking framework, we wanted to acknowledge (1) the specific aspects of PD performance to be assessed, (2) a defined set of benchmarks with which the model’s performance is tested, (3) metrics for measuring and comparing the models to identify strengths and areas of improvement, and (4) avenues for the personalization of the PD procedure.

Since we had a detailed dataset available from both pigs and patients (in terms of solute concentration measurement), we were able to perform the following tasks:(a)Rigorously check if the models were accurate ($C_{error}$ and SSE) and efficient (computational time) overall.(b)Rigorously check to determine how physical complexity helped in detailing the solute kinetics and how that affected the efficiency.(c)Apply TPM for the first time to pigs, thereby enabling the analysis of the differences between pigs and humans in conventional PD and peritoneal membrane characteristics.

Our findings indicate that TPM and WM offer the most accurate predictions for dialysate concentrations of various solutes (including sodium) across both pig and human subjects. This superiority of TPM is attributed to their detailed consideration of solute-specific transport mechanisms, including diffusion and convective transport through different pore sizes, which enhances their predictive power and general applicability. Also, GM and SWM showed good performance, except in the case of sodium. Although not all solutes were available for both pigs and humans, the comparison of four solutes (urea, creatinine, glucose, and sodium) remains valuable in assessing model performance across species under realistic experimental conditions.

TPM has been previously applied to humans [15,16,42], rats [43,44,45], and mice [46]. To the best of our knowledge, this is the first time that TPM has been applied to pigs. Since there are many versions of the TPM [8,10,12,42], including varying definitions of hydraulic conductivity, lymphatic flow rate, and MTAC, we decided to determine these parameters specifically for pigs to enable cross-species comparison (Figure 6). We found that MTACs were generally higher in humans than pigs, which may be due to different peritoneal membrane characteristics, among others, such as the greater thickness of the porcine peritoneum leading to greater diffusion resistance [47]. TPM demonstrated clear accuracy in predicting ultrafiltration in pigs, as it did in humans.

We also explored how well the fitted parameters match the literature/experimentally determined values (parameter plausibility). Plausibility is defined as positive MTAC values and values within ±10% of TPM literature values. Note that we only define plausibility for humans. Here, we combined these measures to analyze the overall model validity in pigs and humans. We found TPM and WM to be the overall best choices for modeling conventional PD. Both models predicted the sodium dip accurately. However, to achieve this prediction, WM fitted MTAC to highly negative values (Figure 6), while TPM fitted the lymphatic flow rate to be unusually high: up to 6 mL/min in pigs and 4 mL/min in humans (Figure 7b). This highlights the need for accurate and frequent volume measurement and representation.

This study highlights a trade-off between model complexity and computational efficiency. While TPM takes a more physiological approach, it suffers from a long computational time. Thus, TPM is not as useful in instant calculations unless fast computation is available to clinicians. On the other hand, WM is time-efficient, while it suffers from a negative sodium MTAC. Despite their time-saving advantages, simplified models (such as GM and SWM) are prone to overfitting or may inaccurately define solute kinetics (underestimation of clearances, negative MTACs). This suggests a need for the cautious interpretation of results and the potential incorporation of regularization techniques or model validation with independent datasets to mitigate these risks. Depending on the clinical scenario, the researcher or clinician has to decide the balance between physical and time complexity. Recent computational advances, such as high-performance parallel computation, machine learning, and neural networks, can be designed to reduce the effective computational time.

The study has several limitations, as listed below:

*Limited intra-abdominal volume data*. We observed a significant challenge across several models to accurately predict sodium transport, which is likely due to complexities associated with sodium sieving and transcellular water flow. This underscores the necessity for the further refinement of the models to incorporate the dynamic interactions between solute transport and water flow. There are more complex fluid models that can be added to the simplistic models like GM and SWM to better describe ultrafiltration [22,48]. We need more volume measurements to decrease the impact of volume error on total error measurements. Precise measurements of residual volumes both before and after dwell can increase the accuracy of the model, as several residual volume measurements were lacking or not reliable. Using a higher glucose concentration—as in human data—creates more favorable conditions for model validations. For TPM, we may also add other physiological phenomena, such as vasodilation, infusion, and a draining procedure.

*Differences in glucose concentration hamper interspecies comparison*. This study’s limitation is that we were not able to find many matched ‘pig-human’ pairs to compare findings because we primarily used glucose 1.36% in pigs (to limit convective flow, allowing for a better evaluation of diffusive processes) and glucose 3.86% in humans (standard clinical protocol).

*Lack of universal benchmarks*. As we navigate the future landscape of in silico medical research, concerted efforts should be directed toward establishing universally accepted benchmarks. These benchmarks will serve as objective, efficient, and reliable tools to evaluate the fundamental properties of PD models, ultimately refining their predictive performance and ensuring optimal patient care. These results contribute to an improved validation of all in silico PD models, particularly for the description of small solute transport that aligns with the specific subject.

## 5. Conclusions

This study presents a comprehensive comparison of six computational models of PD using in vivo data from pigs and humans to predict dialysate concentrations of six solutes (urea, creatinine, phosphate, potassium, glucose, and sodium) in pigs and four solutes (urea, creatinine, glucose, and sodium) in humans, as well as ultrafiltration and the lymphatic flowrate. Through rigorous benchmarking, TPM and WM emerged as the most accurate, although each model’s utility is context-dependent, balancing complexity and computational efficiency. This research underscores the importance of model selection based on the specific requirements of clinical practice or research studies. Furthermore, it highlights the need for continuous model refinement, especially for challenging aspects such as sodium transport, and the potential for personalizing PD treatment through model adaptation to individual physiological characteristics. By advancing our understanding of the predictive performance of PD models, this work contributes significantly to the field of nephrology, offering insights into optimized PD treatment and a deeper comprehension of the PD process. Future directions should be aimed at integrating more complex physiological processes into the models and leveraging emerging computational techniques to enhance predictive accuracy and clinical applicability, ultimately improving personalized patient care in PD therapy.

## Figures and Tables

**Figure 1 toxins-17-00329-f001:**
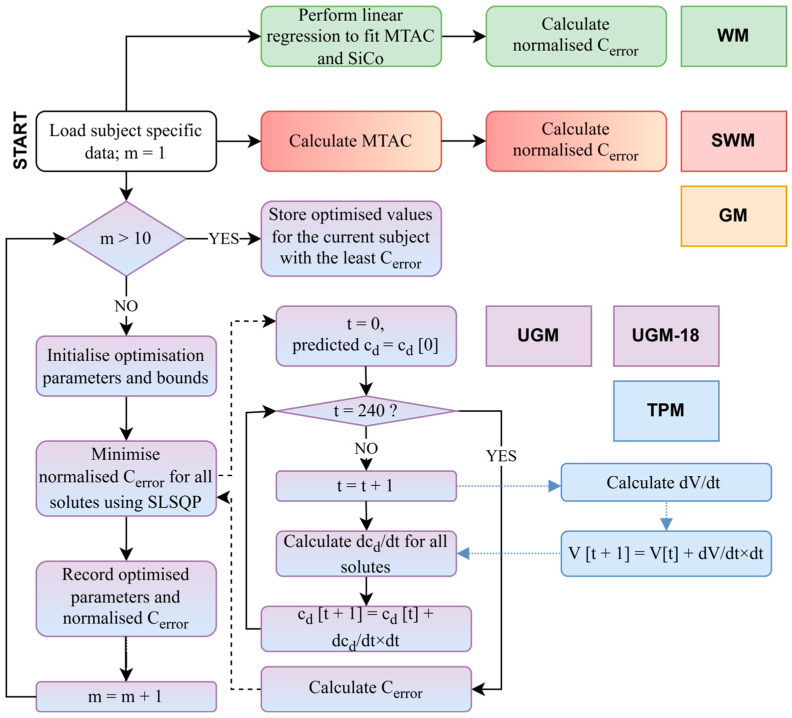
Flowchart explaining the fitting process used for all the models in this paper. (*m* is the iteration number, which varies from 1 to 10, the timestep, *t*, varies between 0 and 240, $c_{d}$ is the dialysate concentration, and V is the intraperitoneal volume.)

**Figure 2 toxins-17-00329-f002:**
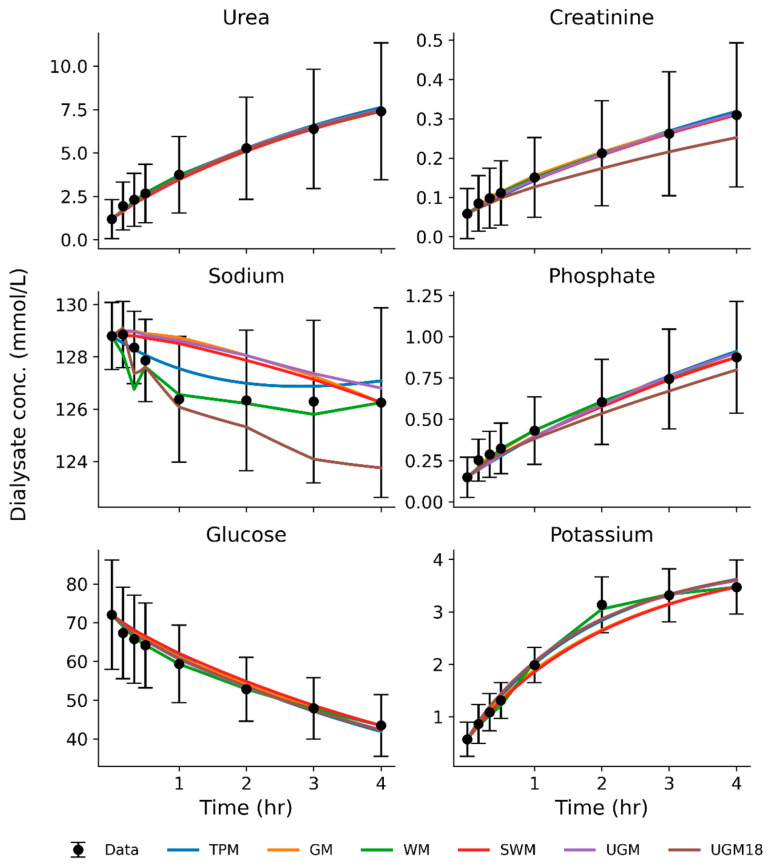
Average predicted dialysate concentration for each model and average measured dialysate concentration (± standard deviation) in 20 sessions in 4 pigs. TPM: three-pore model; GM: Garred model; WM: Waniewski model; SWM: simplified Waniewski model; UGM: unified Graff model; UGM18: fullfit unified Graff model.

**Figure 3 toxins-17-00329-f003:**
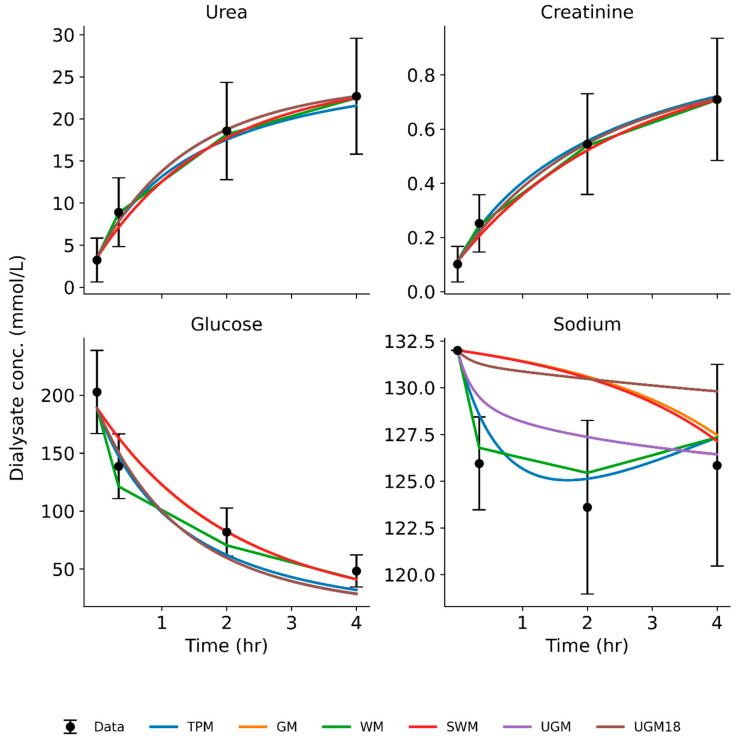
Average predicted dialysate concentration for each model and average measured dialysate concentration (± standard deviation) in 9 dwell sessions in 9 humans. TPM: three-pore model; GM: Garred model; WM: Waniewski model; SWM: simplified Waniewski model; UGM: unified Graff model; UGM18: fullfit unified Graff model.

**Figure 4 toxins-17-00329-f004:**
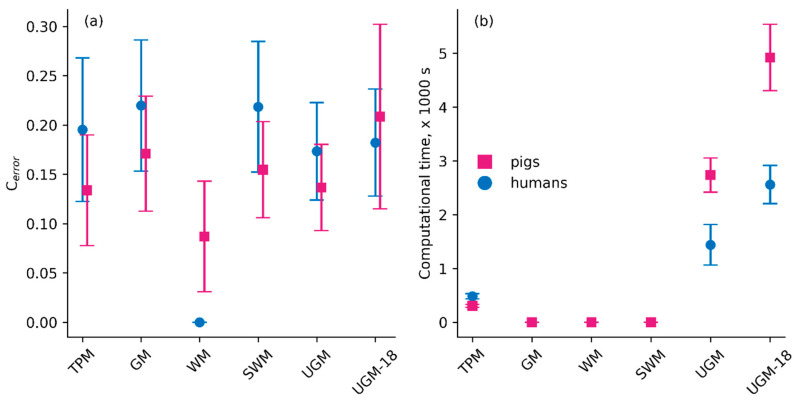
(**a**) Total solute error, $C_{error}$ (mean ± standard deviation), calculated for all solutes for the six models in the entire pig dataset (20 dwells in 4 pigs) and the human dataset (9 dwells in 9 patients). (**b**) Computational time (mean ± standard deviation) for the models. See Figure 1 for a schematic representation of the fitting procedure. TPM: three-pore model; GM: Garred model; WM: Waniewski model; SWM: simplified Waniewski model; UGM: unified Graff model; UGM18: fullfit unified Graff model.

**Figure 5 toxins-17-00329-f005:**
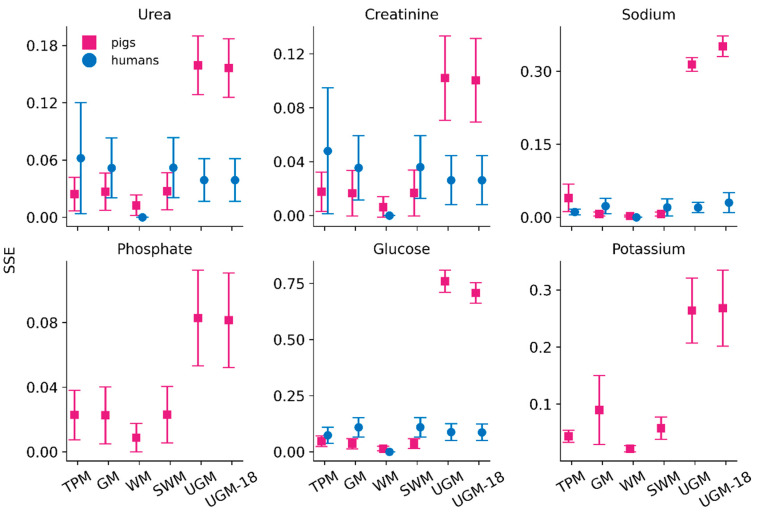
Solute-specific normalized error in concentration determination (SSE) calculated for the six models in pigs (20 dwells in 4 pigs) and humans (9 patient PD dwells). TPM: three-pore model; GM: Garred model; WM: Waniewski model; SWM: simplified Waniewski model; UGM: unified Graff model; UGM18: fullfit unified Graff model.

**Figure 6 toxins-17-00329-f006:**
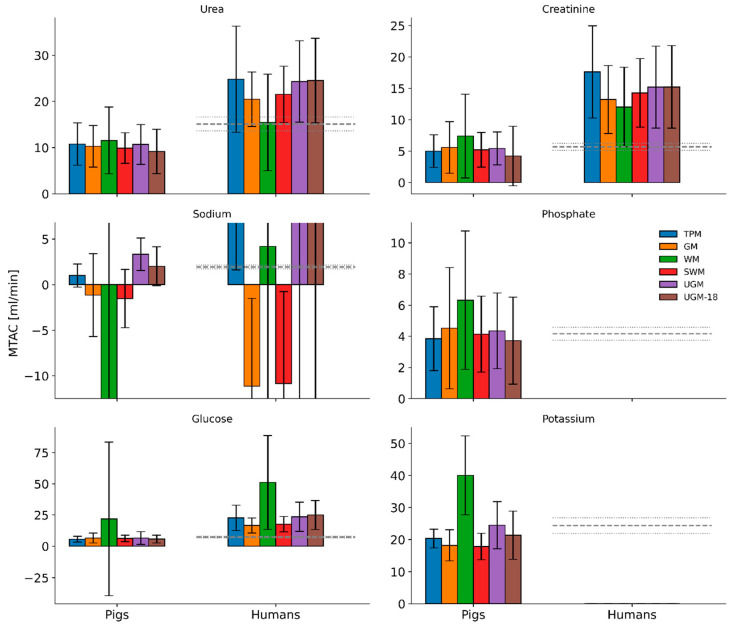
Comparison of model predicted mean solute mass transfer area coefficients (MTACs). TPM: three-pore model; GM: Garred model; WM: Waniewski model; SWM: simplified Waniewski model; UGM: unified Graff model; UGM18: fullfit unified Graff model. The dashed line indicates the literature TPM values for humans [10]. The dotted lines show ±10% of the literature values.

**Figure 7 toxins-17-00329-f007:**
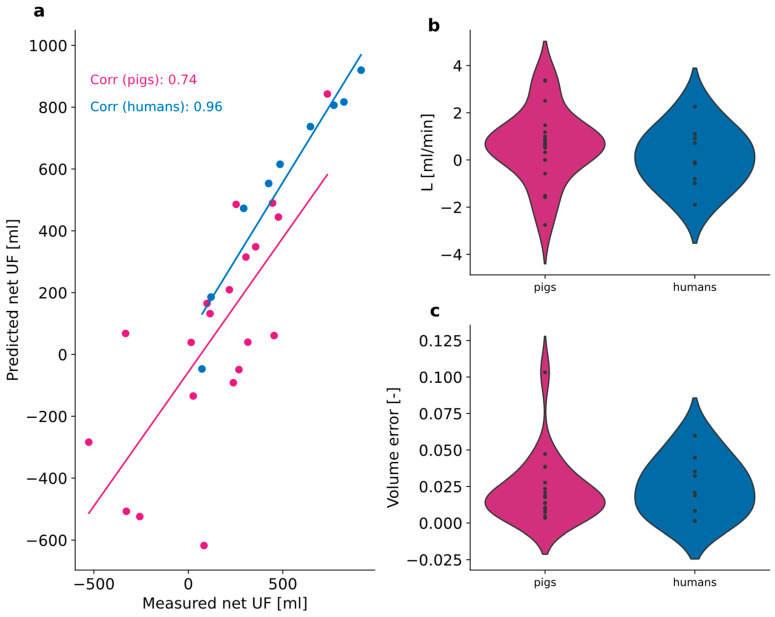
(**a**) Predicted versus measured ultrafiltration, UF, in pigs (*N* = 20) and humans (*N* = 9). (**b**) Lymphatic flow rate, L, predicted in pigs and humans. (**c**) Total volume error, $V_{err}$, in pigs and humans.

**Table 1 toxins-17-00329-t001:** Used PD models and their variations.

Model	Ref	Notes	Number of Fitted Parameters	Clinical Inputs for This Study
No Fluid Model + Linear Solute Model				
Garred model (GM)	[20]	Default model	0	Initial and final dialysate and plasma concentration, Instilled and drained volume
Linear Fluid Model + Non-linear Solute Model				
Graff (UGM)	[23,24,25,26,27,28]	Graff and Fugleberg et al. (1994–1996) established a series of models to fit measured concentrations of sodium, potassium, creatinine, phosphate, glucose, and urea. For the comparison, we only use the best-performing model for the solute and set some parameters according to previous work (see Appendix A).	7 (humans) 11 (pigs)	Dialysate and plasma concentration, Instilled and drained volume
Waniewski model (WM)	[5]	Default model	6 (humans) 12 (pigs)	Dialysate and plasma concentration, Instilled and drained volume
Dynamic Fluid Model + Non-linear Solute Model				
Three-pore model (TPM)	[10]	The three-pore model, originally the two-pore model by Rippe [14], has been developed for continuous flow PD. Here, we have tweaked the model to resemble a typical PD dwell.	5 (humans) 7 (pigs)	Dialysate and plasma concentration, Instilled and drained volume, Residual volume
Variations				
Fullfit Unified Graff Model UGM-18	-	For this variation of the Graff model, we did not restrict any of the parameters to the previously fitted values.	9 (humans) 18 (pigs)	Dialysate and plasma concentration, Instilled and drained volume
Simplified Waniewski Model (SWM)	[35]	For this variation, we used the Garred model with *f* set to 0.5 instead of 0.	0	Initial and final dialysate and plasma concentration, Instilled and drained volume

## Data Availability

The original contributions presented in this study are included in the article/Appendix A. Further inquiries can be directed to the corresponding author(s).

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
