# Peer review of "Comparing Computational Peritoneal Dialysis Models in Pigs and Patients"

_toxins, 2025, doi:10.3390/toxins17070329_

Round 1
Reviewer 1 Report
Comments and Suggestions for Authors
In the manuscript authors focussed on comparing the accuracy of 4 models (plus two variations) both in humans and pigs, since pigs are commonly used for preclinical validation of PD innovations, and authors claim that this is the first time that the three-pore model (TPM) has been applied to pigs. Present work gives crucial insights into which model could be utilized to accurately represent the dynamic interplay of solute transport, paving the way for personalized dialysis care.
Overall manuscript is well written and study has clinical significance towards personalized medical care, therefore, may be accepted for publication.
Author Response
Comments 1:
In the manuscript authors focussed on comparing the accuracy of 4 models (plus two variations) both in humans and pigs, since pigs are commonly used for preclinical validation of PD innovations, and authors claim that this is the first time that the three-pore model (TPM) has been applied to pigs. Present work gives crucial insights into which model could be utilized to accurately represent the dynamic interplay of solute transport, paving the way for personalized dialysis care.
Overall manuscript is well written and study has clinical significance towards personalized medical care, therefore, may be accepted for publication.
Response 1: We sincerely thank the reviewer for their thoughtful and encouraging feedback. We are pleased that the significance of applying and comparing the models—particularly the three-pore model in a preclinical pig setting—was well received.

Reviewer 2 Report
Comments and Suggestions for Authors
In this study the authors different models of PD for removal of toxins in pigs and humans. Despite this several points should be addressed before publishing.
Abstract should be adhere standard abstract writing background, aims, methods, results and conclusions.
Introduction, first sentence is supported with 19 references, please justify. Please, add hint about general methods for toxins removal. Moreover, add advantages and limitations of PD as selective methods for toxins removal. Please add hint about each model of PD selected in this study. Give hint about MTACKs. Please clarify the type of study. Please add the study rationale and specify the names of selected solutes in the study aims.
Each figure and table should be self presented for statical data ( data expression, samples number, significant limit, and p value).
As well as acronyms uses and long sentences or paragraphs without references citation.
Please , discuss the study limitations.
Conclusions, please specify the type of solutes ( urea, glucose, creatinine, etc). studied in this work.
Please, check the manuscript for misuse of acronyms.
Most of the references list are very old please check and correct.
Comments on the Quality of English Language
Please check the manuscript for minor grammar errors and syntax.
Author Response
Comment 1: Abstract should be adhere standard abstract writing background, aims, methods, results and conclusions.
Response 1: We kindly disagree. We have adhered to the instructions from Toxins.
The abstract should be a total of about 200 words maximum. The abstract should be a single paragraph and should follow the style of structured abstracts, but without headings: 1) Background: Place the question addressed in a broad context and highlight the purpose of the study; 2) Methods: Describe briefly the main methods or treatments applied. Include any relevant preregistration numbers, and species and strains of any animals used; 3) Results: Summarize the article's main findings; and 4) Conclusion: Indicate the main conclusions or interpretations.
Comment 2: Introduction, first sentence is supported with 19 references, please justify.
Response 2: We thank the reviewer for the helpful comment. Our initial intent in including 19 references in the first sentence was to offer a comprehensive overview of the range of modeling approaches relevant to our study. These references reflect key models that are either used directly or discussed throughout the paper. However, we agree that concentrating all of them in the opening sentence may affect readability. In response to the comment, we have now redistributed these references to a later section of the introduction, where they are more contextually appropriate and better support the flow of the text.
Comment 3: Please, add hint about general methods for toxins removal.
Response 3: We thank the reviewer for this suggestion which we have included in the introduction:
Dialysis is a life-sustaining therapy that removes waste products and excess fluid from the body in patients with kidney failure to partly replace the kidney function. Hemodialysis is the process of filtering out uremic toxins from the blood is circulated through an extracorporeal circuit along a semipermeable membrane where uremic toxins and excess water are removed by diffusion and ultrafiltration while peritoneal dialysis (PD) is the process of removing of excess water and uremic toxins by introducing a dialysis fluid with osmotic agents in the abdominal cavity of patients.
Comment 4: Moreover, add advantages and limitations of PD as selective methods for toxins removal.
Response 4: We thank the reviewer for the suggestion which we have incorporated in the introduction:
PD offers several benefits over HD, including continuous and gradual removal of waste products—avoiding the sharp fluctuations in solute and fluid levels typical of HD’s intermittent ‘saw-tooth’ pattern. PD does not require vascular access, reducing associated complications, and can easily be performed at home, promoting greater pa-tient autonomy and quality of life. PD better preserves residual kidney function, which is associated with improved patient outcomes, and is generally less expensive. Despite these benefits, PD faces important limitations. Technique survival is limited, with a median duration of approximately 3.7 years, due to complications such as recurrent peritonitis, catheter malfunction, or long-term damage to the peritoneal membrane from exposure to the high-glucose dialysate[1, 2]. Additionally, PD is less effective in clearing toxins compared to HD[3, 4]. As a result, many patients eventually have to transition to HD due to technique failure or inadequate toxin clearance as residual kidney function declines.
Comment 5: Please add hint about each model of PD selected in this study.
Response 5: We thank the reviewer for the suggestion which we have included as follows in the introduction:
The PD models of Kallen[20][15], Miller[16][21] and Henderson[17][22] are two-compartmental models, which consider the peritoneal membrane to be homogeneously porous and the solute transfer across the membrane purely diffusive. However, as our understanding of the peritoneal membrane has increased, we now know that there is also convective solute transfer and that solutes are transferred depending on size and charge, thus indicating a heteroporous network. Such heteroporous network was incorporated in the simplified models of Garred[4] and Waniewski[18, 19]. We also know that there are aquaporin channels for water transfer[23] and that the lymphatic system is important for the removal of fluid and macromolecules[24-28]. Consequently, recent PD models like the solute-specific models of Graff and Fugleberg[20-25], the three-pore model (TPM) of Öberg[26], the empirical model of Gotch[27] and the distributed model of Flessner[28, 29] have incorporated this complexity, underscoring the importance of diffusion, convection and lymphatics.
Comment 6: Give hint about MTACs.
Response 6: The following sentence has been added to the manuscript’s introduction section.
The MTAC, which represents the maximal diffusive clearance, reflects the capacity for diffusion across the peritoneal membrane and can be calculated based on solute transport dynamics.
Comment 7: Please clarify the type of study. Please add the study rationale and specify the names of selected solutes in the study aims.
Response 7: We thank the reviewer for this comment. We have now added the type of study and the names of the solutes in the introduction.
This is a benchmarking modeling study aimed at evaluating how accurately the models predict the dialysate concentrations and mass transfer area coefficients (MTACs) of six solutes (urea, creatinine, phosphate, potassium, glucose, sodium) in pigs and four solutes (urea, creatinine, glucose, sodium) in humans.
The study rationale has already been discussed in this paragraph in the introduction:
However, a rigorous comparison of the accuracy of various models is needed, since an accurate model may contribute to the optimization and personalization of this life-saving therapy and may be used to optimally design PD innovations that are currently under development, such as sorbent-assisted continuous flow PD [30, 31].
Comment 8: Each figure and table should be self presented for statical data ( data expression, samples number, significant limit, and p value).
Response 8: We thank the reviewer for this pertinent remark. We have revised the figures and tables to be self-explanatory, including e.g. a description of the sample number, units and plotted metrics.
Comment 9: As well as acronyms uses and long sentences or paragraphs without references citation.
Response 9: We thank the reviewer. We thoroughly checked and revised the manuscript accordingly.
Comment 10: Please , discuss the study limitations.
Response 10: We thank the reviewer for the comment. The study limitations are already in the discussion section of the manuscript as copied below:
The study has several limitations as listed below:
Limited intra-abdominal volume data: We observed a significant challenge across several models to accurately predict sodium transport, likely due to complexities associated with sodium sieving and transcellular water flow. This underscores the necessity for further refinement of models to incorporate the dynamic interactions between solute transport and water flow. There are more complex fluid models that can be added to the simplistic models like GM and SWM to better describe ultrafiltration[52,53]. We need more volume measurements to decrease the impact of volume error on total error measurements. Precise measurements of residual volumes both be-fore and after dwell can increase the accuracy of the model as currently several residual volume measurements were lacking or not reliable. Higher glucose concentration used – as in human data creates more favorable conditions for model validations. For TPM, we may also add other physiological phenomena such as vasodilation, infusion and draining procedure.
Differences in glucose concentration hamper interspecies comparison. The study’s limitation is that we were not able to find many matched ‘pig-human’ pairs to compare findings because we primarily used glucose 1.36% in pigs (to limit convective flow al-lowing better evaluation of diffusive processes) and 3.86% in humans (standard clinical protocol).
Lack of universal benchmarks. As we navigate the future landscape of in silico medical research, concerted efforts should be directed toward establishing universally accepted benchmarks. These benchmarks will serve as objective, efficient, and reliable tools to evaluate the fundamental properties of PD models, ultimately refining their predictive performance and ensuring optimal patient care. These results contribute to an improved validation of all in silico PD models, particularly for description of small solute transport that align with the specific subject.
Comment 11: Conclusions, please specify the type of solutes ( urea, glucose, creatinine, etc). studied in this work.
Response 11: We thank the reviewer for the comment. This has been rectified in the text as follows in the conclusion.
This study presents a comprehensive comparison of six computational models of PD using in vivo data from pigs and humans to predict dialysate concentrations of six solutes (urea, creatinine, phosphate, potassium, glucose, sodium) in pigs and four solutes (urea, creatinine, glucose, sodium) in humans, as well as ultrafiltration and lymphatic flowrate.
Comment 12: Please, check the manuscript for misuse of acronyms.
Response 12: We have checked the manuscript as closely as possible to make sure that all acronyms are fully described in the manuscript as well as glossary.
Comment 13: Most of the references list are very old please check and correct.
Response 13: We thank the reviewer for this observation. The older references were included intentionally to illustrate the historical evolution and increasing complexity of peritoneal dialysis (PD) modeling over time. However, in response to the comment, we have carefully reviewed the reference list, removed redundant older citations, and updated the manuscript by including more recent and relevant references to balance older as well as newer models in our comparison.

Reviewer 3 Report
Comments and Suggestions for Authors
The present manuscript „Comparing Computational Peritoneal Dialysis Models in Pigs and Patients” tested six computational models for their predictability of in vivo PD data. Such modeling is important to reduce the need of clinical and animal studies. The manuscript is well written, and the results are clearly presented.
Here are some comments which may help to further improve the manuscript:
- The introduction mentions several models (e.g. Kallen, Miller, Henderson) but not the investigated ones (UGM, TPM, GM, WM). This could be added to the introduction and shortly described why these ones were selected.
- The manufacturers of the products could be added (e.g. Physioneal)
- Line 91: SPA should be explained
- Line 121: MTAC should be explained
- An explanation could be added why not the same solutes for pigs and humans were investigated
Author Response
The present manuscript „Comparing Computational Peritoneal Dialysis Models in Pigs and Patients” tested six computational models for their predictability of in vivo PD data. Such modeling is important to reduce the need of clinical and animal studies. The manuscript is well written, and the results are clearly presented.
Here are some comments which may help to further improve the manuscript:
Comment 1: The introduction mentions several models (e.g. Kallen, Miller, Henderson) but not the investigated ones (UGM, TPM, GM, WM). This could be added to the introduction and shortly described why these ones were selected.
Response 1: We thank the reviewer for this comment. We have made sure to include the used models in our introduction text as follows.
The PD models of Kallen[20][15], Miller[16][21] and Henderson[17][22] are two-compartmental models, which consider the peritoneal membrane to be homogeneously porous and the solute transfer across the membrane purely diffusive. However, as our understanding of the peritoneal membrane has increased, we now know that there is also convective solute transfer and that solutes are transferred depending on size and charge, thus indicating a heteroporous network. Such heteroporous network was incorporated in the simplified models of Garred[4] and Waniewski[18, 19]. We also know that there are aquaporin channels for water transfer[23] and that the lymphatic system is important for the removal of fluid and macromolecules[24-28]. Consequently, recent PD models like the solute-specific models of Graff and Fugleberg[20-25], the three-pore model (TPM) of Öberg[26], the empirical model of Gotch[27] and the distributed model of Flessner[28, 29] have incorporated this complexity, underscoring the importance of diffusion, convection and lymphatics.
Comment 2: The manufacturers of the products could be added (e.g. Physioneal)
Response 2: We thank the reviewer for this helpful suggestion and agree that including the manufacturers is important for clarity and reproducibility. Upon reviewing the manuscript, we noted that the names of the manufacturers have already been provided in Section 2.1, specifically on lines 105, 109, 114, and 121. We hope this level of detail sufficiently addresses the reviewer’s concern.
Comment 3: Line 91: SPA should be explained
Response 3: We thank the reviewer for pointing this out. We have now clarified the abbreviation in the text. The sentence has been revised to:
The instilled and drained intraperitoneal volume were measured along with blue dextran measurements for intermediate volume data. In short, blue dextran (Sigma Aldrich; D5751-10g, 2000 kDa) was dissolved in fresh Physioneal 35 1.36% and added to the dialysate for the fresh fill at the start of the standard peritoneal assessment to achieve a concentration of 0.49 ± 0.14 g/L (n=21).
Comment 4: Line 121: MTAC should be explained
Response 4: We thank the reviewer for this comment. We do not find the acronym in line 121 in the submitted version of the manuscript. However, it is present in line 58 (written out in full along with the abbreviation). We have also included a description of MTAC in the revised manuscript.
The MTAC, which represents the maximal diffusive clearance, reflects the capacity for diffusion across the peritoneal membrane and can be calculated based on solute transport dynamics.
Comment 5: An explanation could be added why not the same solutes for pigs and humans were investigated.
Response 5: We thank the reviewer for this observation. As the computational study was conducted retrospectively, the selection of solutes was based on the available data from previously performed experiments. Not all solutes were measured consistently across both species, which limited our ability to include an identical set of solutes for pigs and humans. We have now clarified this point in the manuscript to ensure transparency regarding data availability.
Methods
All the data used to compare the models have been procured retrospectively. As this study was conducted retrospectively, the set of solutes analyzed in pigs and humans reflects the measurements available from prior experimental protocols; six solutes for pigs (urea, creatinine, phosphate, potassium, glucose, sodium) and four solutes for humans (urea, creatinine, glucose, sodium). Consequently, not all solutes were available for modeling in both species.
Discussion
Although not all solutes were available for both pigs and humans, the comparison for four solutes (urea, creatinine, glucose and sodium) remains valuable in assessing model performance across species under realistic experimental conditions.
